# Effect of a Physical Education Teacher’s Autonomy Support on Self-Esteem in Secondary-School Students: The Mediating Role of Emotional Intelligence

**DOI:** 10.3390/children10101690

**Published:** 2023-10-16

**Authors:** Antonio Granero-Gallegos, Manuel Gómez-López, David Manzano-Sánchez

**Affiliations:** 1Department of Education, University of Almeria, 04120 Almeria, Spain; agranero@ual.es; 2Health Research Centre, University of Almeria, 04120 Almeria, Spain; 3Department of Physical Activity and Sport, Faculty of Sports Sciences, University of Murcia, 30720 Murcia, Spain; 4Department of Didactics of Musical, Plastic and Corporal Expression, Faculty of Education and Psychology, University of Extremadura, 06006 Badajoz, Spain; davidms@unex.es

**Keywords:** secondary education, self-esteem, emotional attention, emotional clarity, emotional repair

## Abstract

The purpose of this study was to analyse the effect of emotional intelligence between the perception of autonomy support from physical education teachers and self-esteem (positive and negative) in secondary-school physical education students. The study design was observational, descriptive, and cross-sectional. In total, 1069 secondary-school physical education students participated (*Mage* = 14.55; *SD* = 1.54) (51.2% female; 48.8% male). The following scales were used: The Learning Climate Questionnaire adapted to Physical Education (i.e., autonomy support), the Trait Meta-Mood Scale-24 (i.e., emotional attention, emotional clarity, and emotional repair), and the Rosenberg Self-Esteem Scale (positive self-esteem, negative self-esteem). A structural equation model was performed with the latent variables controlled by age, sex, and the students’ educational centre. The main results indicate that the explained variance was 37% for positive self-esteem and 26% for negative self-esteem. In addition, autonomy support directly predicts emotional intelligence (*p* < 0.05) and positive self-esteem (*p* < 0.001). On the other hand, all indirect effects of autonomy support on self-esteem across emotional intelligence were significant at *p* < 0.001. Finally, emotional clarity and emotional repair had a mediating effect on self-esteem, and it improves the total effect of autonomy support on positive self-esteem with values of β = 0.14 and β = 0.19, respectively, and a value of β = −0.07 and β = −0.06 for negative self-esteem. The findings reveal the necessity to improve emotional clarity and emotional repair in secondary-school students in improving positive self-esteem through the perception of autonomy support from the physical education teacher.

## 1. Introduction

Adolescence is a period of transition during which young people experience a multitude of changes at the cognitive, social, and psychological levels [1]. All these changes can influence a variety of factors, one of the most prominent being self-esteem. High positive self-esteem plays an important role in a person’s development, affecting emotional well-being and the ability to overcome adversity and to resist social pressure [2], and even the ability to have more satisfying social relationships, confidence, and satisfaction in their own life [3]. In contrast, negative self-esteem is related to negative consequences, such as depression or anxiety, as indicated by the American Psychological Association [4]. Throughout a person’s developmental process, education plays a fundamental role. Physical education (PE) in particular is a subject that can have a positive impact on a student’s self-esteem, especially if decision making is encouraged through teaching–learning situations where autonomy [5,6,7] and the perception of competence [8] are favoured. Similarly, emotional intelligence (EI), understood as the ability to recognize and manage emotions, is closely related to positive self-esteem, with PE being a context in which various research studies have been carried out that support the importance of the subject in developing this ability—e.g., [9]. Therefore, studying potential antecedents of positive self-esteem in students during adolescence is of particular interest, as indicated above, along with the support for autonomy on the part of the teacher and the ability to manage and handle one’s own emotions on the part of the student. These aspects have been related to self-esteem, so analysing them could help to better understand the development of positive self-esteem, in this case, in the specific context of PE.

### 1.1. Autonomy Support (AS)

Autonomy is understood as the ability to decide and act for oneself, without external restrictions and in accordance with one’s own values and objectives [10]. The importance of autonomy in students has attracted the attention of educators [11], being a fundamental aspect of the mission and vision of schools [12]. Since the teacher is a fundamental element in creating an adequate educational environment [13], support for autonomy—understood as instruction through a way of teaching that appreciates, supports, and vitalizes needs [14]—is recognized as a facilitator of the students’ autonomous behaviours [15]. This is backed up with Self-Determination Theory (SDT; [16,17]). SDT is a theory of human motivation that aims to predict and explain people’s behaviours in different contexts, indicating that when there is a trigger (such as the perception of support for autonomy), it can create new behaviours and conduct in people (e.g., autonomous behaviours) through various motivational processes. More specifically in the educational field, and following Vasconcellos et al. [18], it can be said that adequate support for autonomy correlates with more autonomous motivation and, along with it, various behaviours and conduct that enable better social adaptation in children and adolescents. Along these lines, Sun et al. [19] highlight the importance of SDT in relation to the students’ psychomotor and affective domains, leading to improved self-esteem [20,21]. In addition, Oh and Cho [22] recently reported on the mediating role of EI between support for autonomy and cognitive variables (i.e., interruption intention) although this was carried out in the sports field. For this reason, it would also be interesting to study these relationships (AS- > EI- > self-esteem) in the educational field.

### 1.2. Emotional Intelligence

In recent years, educational research has focused on narrowing down the personal skills that allow people to improve emotional well-being, with the study of EI being one of the research lines that most focuses on analysing personal differences [23]. Specifically, EI refers to the ability to define one’s own feelings and those of others, motivate oneself, and manage emotions in intra- and interpersonal relationships [24]. According to Salovey et al. [25], EI is composed of three sub-dimensions: (i) attention (EA), or the ability to perceive one’s own emotions and those of others (emotional attention); (ii) clarity (EC), or the ability to understand emotional information (how emotions combine and progress over time) and to understand emotional meanings (emotional clarity); and (iii) repair (ER), or the skill to change feelings and those of others as well as to promote understanding and personal growth (emotional repair). The importance of studying EI in children and adolescents is continuously growing [26,27,28,29] and some authors have recognized that among the factors influencing teaching success are EI and self-esteem [30]. In addition, according to Cheung et al. [31], EI can be a determining aspect of self-esteem. Recent research in the educational field has linked the dimensions of EI (i.e., emotional attention, emotional clarity, and emotional repair) with self-esteem [32] and even shown EI to be a predictor of self-esteem [33]. Therefore, to advance in this field of study, one might ask how the student manages support for autonomy in PE classes according to the various factors of EI and how this can predict improved self-esteem in these students.

### 1.3. Self-Esteem and Teacher Support

Self-esteem is the set of opinions that an individual has about themself and the expectations related to their approval or disapproval [24], with adequate self-esteem being one of the most valued personal resources a person can have [34], in addition to being of transcendental importance in the educational system [35,36]. According to Rosenberg [37], a person with adequate self-esteem is someone who considers themselves valuable, capable of recognizing their mistakes, and apt to value their merits. Following this author’s theory, self-esteem is based on two aspects: (i) reflective assessments and (ii) social comparisons. Regarding these, Rosenberg indicated that human communication depends on being able to understand the perspective of other people whereas social comparisons emphasize that self-esteem is also a consequence of subjects comparing themselves with each other and making positive (positive self-esteem) or negative (negative self-esteem) self-assessments, all of which depend on assessing the social group. The literature has shown the benefits of adequate positive self-esteem in the personal, cognitive, and social spheres —e.g., [38,39]. Similarly, negative self-esteem, especially in adolescents, is a predictor of deteriorating mental health [40] or a deterioration in social relationships [3]. The influence of people who are close to the subject, such as family members and teachers, plays an important role in the student’s self-esteem [41,42]. Indeed, recent studies, such as the one by Valero-Valenzuela et al. [6], demonstrate that support for autonomy, as opposed to a controlling style on the part of the teacher in PE classes, enables increased self-esteem in students by providing them with the opportunity to make decisions while performing learning tasks.

### 1.4. The Present Study

It has been widely described in the scientific literature that greater support for autonomy can be a predictor of positive consequences through the development of EI [22] and that EI can be decisive in obtaining greater self-esteem [31,32,33,43]. However, to the best of our knowledge, there is no research in the literature that has analysed the relationship of the three variables together (i.e., perception of AS from the teacher, EI, and self-esteem). Furthermore, although some studies have been carried out in the PE field—e.g., [44]—the relationship between support for autonomy from the teacher and physical self-esteem has been investigated but not self-esteem in general. This study analyses this relationship within PE classes in secondary-school students using a structural equations analysis, and it represents an interesting contribution to the literature and an advance in scientific knowledge. 

Therefore, the objective of this study was to analyse the effect of support for autonomy from the PE teacher on the secondary physical education students’ self-esteem (positive and negative), while analysing the mediating role of EI (EA, ER, and EC). Taking into account the reviewed literature, a hypothesized model is considered (see Figure 1) with the following hypotheses: H1—AS will be positively related to the three dimensions of EI; H2—the three dimensions of EI will be positively related to positive self-esteem, and negatively related to negative self-esteem; H3—AS will act as a direct and positive predictor of positive self-esteem and a direct and negative predictor of negative self-esteem; and H4—the three dimensions of EI will act as positive mediators between AS and the two dimensions of self-esteem.

## 2. Materials and Methods

### 2.1. Design and Participants 

This research study followed a descriptive, observational, cross-sectional, and non-randomized design. The students who participated were enrolled in one of seven public Secondary Education Institutes (IES) in the Region of Murcia (Spain). To take part in this study, the following inclusion criteria were specified: (i) to be enrolled in a secondary education course; (ii) to be taking the PE course and to attend regularly; (iii) to provide informed consent signed by a parent/guardian to participate in the study; and (iv) to complete the data collection form with the different scales. It should be noted that an a priori analysis was carried out of the required sample size, considering a structural equation model (SEM) approach with 6 latent variables and 48 observable variables to fulfil the study objective. It was estimated that a minimum of 1019 students were needed for an effect size of f^2^ = 0.155, having a statistical power of 0.95% and a significance level of α = 0.05. In the end, 1069 secondary school students (51.2% female; 48.8% male) participated in this research, all coming from one of seven public secondary schools located in both urban and rural areas within the Region of Murcia (Spain). Their age ranged from 12 to 17 years (*M* = 14.55; *SD* = 1.54). The classes were mixed (boys and girls) and all the students took PE as a compulsory subject (2 h per week). The data were collected during the second semester of the school year in seven different secondary education centres. The contents of PE classes are normatively regulated with RD 1105/2014, but each educational centre has autonomy to organize teaching differently, both in the course and at the time of the course in which it is taught. Likewise, each teacher has the autonomy to choose the methodology that they believe is most appropriate according to the contents and characteristics of their pupils.

### 2.2. Instruments

Autonomy support. The Spanish validated version of the Learning Climate Questionnaire (LCQ- PE) by Granero-Gallegos et al. [45], from the original scale by Williams and Deci [46], was administered to the secondary-education PE classes. This instrument is composed of 14 items (e.g., “I feel that my physical education teacher provides me with choices and options”) grouped in a dimension that measures the perception of autonomy support given by the teacher. A Likert scale ranging from 1 (strongly disagree) to 7 (strongly agree) was used for the responses. In the present study, the value of Omega was 0.95.

Emotional Intelligence. The Trait Meta-Mood Scale-24 (TMMS-24), validated in Spanish by Fernández-Berrocal et al. [47], was also administered. This instrument is composed of 24 items and is an indicator of the levels of perceived EI. It consists of three dimensions grouped into eight items each: *emotional attention* (EA), which refers to the degree to which people pay attention to their feelings; *emotional clarity* (EC), which refers to how people perceive their feelings; and *emotional repair* (ER), which refers to how each person believes in their ability to interrupt negative emotional states and prolong positive ones. A Likert scale ranging from 1 (strongly disagree) to 5 (strongly agree) was used for the responses. In the present study, the value of Omega was 0.88 (EA); ω = 0.85 (EC); and ω =.83 (emotional repair).

*Self-Esteem.* The Rosenberg Self-Esteem Scale (RSS), validated by Atienza et al. [48], was likewise administered. This scale is composed of 10 items grouped into two dimensions of 5 items each: *positive self-esteem*, which evaluates the level of self-confidence or personal assessment, and *negative self-esteem*, which evaluates the level of self-contempt or personal devaluation. A Likert scale ranging from 1 (strongly disagree) to 4 (strongly agree) was used for the responses. The Omega value was ω = 0.80 and ω = 0.75 to positive self-esteem and negative self-esteem, respectively. 

### 2.3. Procedure

After authorization to carry out the study from the IES management, the PE teaching staff who teach the subject were contacted and the students were informed of the research objectives and its relevance, their rights as participants, how to answer the questionnaire, the anonymity of the responses, and that these would not affect their subject grades. They were also informed that they could cease participation in the study at any time. The data were collected in person by one of the researchers. All the participants included in the research gave their consent before participating. The study was conducted in accordance with the Declaration of Helsinki and approved by the Ethics Committee of the University of Murcia (Ref: 4447/2023). 

### 2.4. Statistical Analysis

Descriptives and correlations between variables were determined using SPSS v.29 software. The McDonald’s Omega coefficient (ω) was estimated as a measure of internal consistency for each dimension included in the research, taking into account that values >0.70 indicate acceptable reliability [49]. Next, following Kline [50], a two-step latent variable SEM was calculated with AMOS v.29 software to study the predictive predictions between the perception of the PE teacher’s support for autonomy and the student’s self-esteem, analysing the mediating role of EI. As a first step (the measurement model), the robustness of the bidirectional relationships between the SEM variables was evaluated. As a second step, the predictive effects between the dimensions were studied. The SEM was controlled using the participating students’ age, sex, and educational centre. Given that the multivariate normality assumption was violated (Mardia’s coefficient = 82.51; *p* < 0.001), the analysis was carried out using the maximum likelihood method and the 5000-iteration *bootstrapping* procedure [50]. The SEM fit was evaluated considering the chi-square ratio and degrees of freedom (χ^2^/df) values, with <5.0 being acceptable; CFI (*Comparative Fit Index*) and TLI (*Tucker–Lewis Index*) values, with >0.90 being acceptable (and >0.95 being excellent); and RMSEA (*Root Mean Square Error of Approximation*) and SRMR (*Standardized Root Mean Square Residual*) values, with <0.06 being excellent [51,52]. The proposal of Shrout and Bolger [53] was addressed in the analysis of direct and indirect effects. The indirect effects (i.e., mediated) and their 95%CI were calculated with the *bootstrapping* technique, considering the indirect effect (*p* < 0.05) as significant if its 95%CI did not include the zero value. Finally, to achieve better results interpretation, the total explained variance (R^2^) was considered as a measure of the effect size (ES) [54], in such a way that values <0.02, close to 0.13, and >0.26 were considered small, medium, or large, respectively, in accordance with Cohen [55]. The CI (95%) of R^2^ was also determined to ensure that no value was less than the minimum required for its interpretation (0.02).

## 3. Results

### 3.1. Preliminary Results

Correlations and a descriptive analysis between the different variables are shown in Table 1.

### 3.2. Main Results

The SEM showed an acceptable fit in step 1: χ^2^/df = 2.51, *p* < 0.001; CFI = 0.96; TLI = 0.95; RMSEA = 0.038 (90%CI = 0.034; 0.041; *p*_close_ = 1.000); SRMR = 0.036. In step 2, the model had similar and acceptable goodness-of-fit indices: χ^2^/df = 2.51, *p* < 0.001; CFI = 0.96; TLI = 0.95; RMSEA = 0.038 (90%CI = 0.034; 0.041; *p*_close_ = 1.000); SRMR = 0.036. The SEM was controlled using the participants’ age, sex, and educational centre. The explained variance reached 37% for positive self-esteem and 26% for negative self-esteem (Figure 2). The relationships between the perception of the PE teacher’s support for autonomy, EI (i.e., EA, EC, ER), and self-esteem (i.e., positive and negative) can be seen in Figure 2 and Table 2.

As shown in the SEM, AS directly and positively predicts EA (*p* = 0.018), EC (*p* = 0.009), ER (*p* = 0.018), and positive self-esteem (*p* < 0.001), while the relationship is negative with negative self-esteem (*p* = 0.020). EA is directly and positively related to negative self-esteem (*p* = 0.009), while the direct relationship is negative with positive self-esteem (*p* = 0.009). Both EC (*p* = 0.014) and ER (*p* = 0.005) directly and positively predict positive self-esteem, while the prediction is negative with negative self-esteem (EC, *p* = 0.009; ER, *p* = 0.014). On the other hand, it should be noted that all indirect effects of AS on self-esteem (positive and negative) across the three dimensions of EI were statistically significant (*p* < *0*.001). However, what is most relevant about the mediating effect of EI on self-esteem is that it improves the total effects of AS on positive self-esteem, both through EC (β = 0.14; *p* < 0.001), and ER (β = 0.19, *p* < 0.001), while also improving the effects on negative self-esteem by decreasing it, also through EC (β = −0.07, *p* < 0.001) and ER (β= −0.06, *p* < 0.001). In the SEM, the total indirect effect of AS on positive self-esteem was β = 0.11 (95%CI = 0.07, 0.15; *p* = 0.008) and on negative self-esteem, it was β = −0.07 (95%CI= −0.11, −0.04; *p* = 0.006). Finally, Figure 2 shows the CI(95%) of R^2^, confirming that they can be considered as measures of the ES.

## 4. Discussion

The objective of this study was to verify the effect of AS from PE teachers on the self-esteem of their secondary-school students by analysing the mediating role of EI. The main results showed that AS plays an important role as a predictor of EI and the most relevant of the mediating effects of EI is that it improves the overall effects of AS on positive self-esteem, both through EC and ER, while also improving the effects on negative self-esteem by decreasing it, again through EC and ER.

The first hypothesis (H1) proposed that support for autonomy would be positively related to the three dimensions of EI (i.e., EA, EC, and ER). First, it should be noted that the results show that support for autonomy directly and positively predicts these three dimensions of EI, so H1 is met. The obtained results corroborate the findings of previous research, both in the academic field with samples of young students in primary [56] and secondary education [57], and in the sport [22] and family context [58]. The importance of students perceiving their teacher’s support for autonomy in developing their EI has already been highlighted in previous studies—e.g., [59]—which states that the highest values of EI, in its three dimensions, correspond to students with a greater perception of autonomy on the part of external agents.

For the second hypothesis (H2), it was proposed that the three dimensions of EI would be positively related to positive self-esteem, and negatively related to negative self-esteem. The results partially corroborate this hypothesis, given that EA negatively predicts positive self-esteem and positively predicts negative self-esteem. These findings follow the line of study conducted by Rey et al. [43], which showed that EC and ER do have a positive effect on self-esteem, but, on the other hand, that EA was not related to self-esteem. This is not surprising given that, as indicated by García-Linares et al. [59], the relationship of the EA dimension with other variables shows contradictory results, probably because attention to feelings can play a different role from those of EC and ER if adequate psychosocial adjustment is verified [60]. These results are similar to those of Ruvalcaba-Romero et al. [61], in which the predictive power of positive self-esteem was achieved with ER and EC. This aspect must be taken into consideration in PE classes since, although support for autonomy predicts greater EA in students, this attention does not mean it will increase positive self-esteem, but rather that it increases negative self-esteem. That is why an excess of EA given to the student can be negative; instead, what we must seek as teachers is to increase the clarity and repair of emotions in students via support for autonomy in order to develop their positive self-esteem in PE classes. A possible explanation for this may be that people who receive higher EA may need more external support. As external incentives, these can trigger a reduction in more internal motivation and personal well-being, with a consequent reduction in positive self-esteem, by seeking external approval and not focusing on personal approval. Recent studies carried out with schoolchildren during COVID-19 highlight the importance of managing emotions at the intrapersonal level and adaptability to improve cognitive variables such as life satisfaction [27] or resilience [29].

The third hypothesis (H3) proposed that support for autonomy would act as a direct and positive predictor of positive self-esteem and a negative predictor of negative self-esteem. The results of our study corroborate that H3 is met. Along these lines, Hein and Caune [20] found in a study conducted on adolescent students that support for autonomy predicts students’ physical self-esteem using motivation as a mediator. At the same time, it should be noted that various research studies have proven the importance of the teacher’s role on students’ self-esteem. For example, in the field of PE, the research conducted by Hein et al. [44], with students from various countries, found that support for autonomy allows for greater physical self-esteem. On the other hand, the role of AS has also been highlighted in the sports field, as evidenced with the systematic review with a meta-analysis carried out by Mossman et al. [62], where the positive effect of support for autonomy on the self-esteem of athletes is highlighted. These findings highlight the need to carry out more studies in the field of education, where support for autonomy is analysed and favoured in order to improve students’ positive self-esteem. 

Finally, the fourth hypothesis (H4) proposed that the three dimensions of EI would act as positive mediators between the support for autonomy and the two dimensions of self-esteem. In this case, the EA acted indirectly in a positive way with negative self-esteem, unlike EC and ER, finding the opposite results for positive self-esteem, which in all cases was significant. As mentioned before, it is an expected result given that the psychological adjustment of people in search of well-being seems to be found in profiles of people with high EI but where the EA variable is usually smaller [63]. As positive self-esteem is an essential variable in psychological and emotional well-being [64], studies such as that by Cheung et al. [31] must be highlighted, which indicate that high EI allows greater self-esteem. This was particularly shown in the study conducted by Ruvalcaba-Romero et al. [61], which states that it is EC and ER especially that act on positive self-esteem. Although the profiles differ according to the three EI dimensions, the results reflect that the profile with greater EA compared to EC and ER has lower self-esteem [32].

### 4.1. Limitations and Future Lines of Study

This research has a series of limitations and strengths that need to be pointed out to understand their characteristics and to indicate the potential for future studies: (i) being a descriptive study, reality cannot be changed to achieve more adaptive results from the study sample; (ii) like any study using administered questionnaires, the responses may include a possible social desirability bias that influences the results; (iii) the sample is non-randomized; and (iv) this study did not take into account the contents that the students were working on at the time of data collection, nor the methodology used by the teacher when teaching PE contents. As its strengths, we can highlight the following: (i) the sample size; (ii) the type of analysis carried out with latent variables; (iii) the proposed mediation analysis that has not (to date) been explored in the literature; and (iv) that general self-esteem was investigated in its two dimensions, positive and negative. Finally, as potential future lines of study, the model could be tested with other samples in the educational field together with other models that include a new consequence variable based on self-esteem. Furthermore, based on the results of the direct effects of the gender and age variables on EI, as well as the total indirect effects (through the EI variables, whose direct effects are the significant ones), future studies should focus on studying these differences in this population. Likewise, controlled interventions could be proposed in which the assignment of autonomy and responsibility by the teacher would enable an improvement in the students’ perception of autonomy. Another option would be to develop specific programs that favour support for self-esteem and the development of EI, where the teacher acts as a mediator, always from the standpoint of encouraging support for students rather than control over them.

### 4.2. Practical Implications

This research is especially useful in verifying the important role played by teachers in the teaching–learning process. The teacher’s support for autonomy and the search for strategies that promote such autonomy are especially relevant given that the development of EI, and self-esteem (directly and indirectly), depends in part on such support. In this case, we verified the relevance of teacher support, but other studies, such as those indicated above, also verified it in other contexts, for example, with coaches or parents [22,58]. Teachers should use strategies to promote such support for autonomy with diverse interventions, as are explained in various studies [65,66,67]. In turn, EA is an EI variable that must be treated with caution since an excess of it can lead to a reduction in the self-esteem we wish to encourage. The use of strategies developing programs for intra- and interpersonal EI work can lead to good results, both at the university level [68] and with children [69], as they emphasize knowing oneself and knowing how to act with others. In this same sense, PE classes that allow students to make decisions autonomously, thanks to the support of their teachers, can have a positive impact on their self-esteem, given that those who perceive support for autonomy from the teachers are more motivated, having greater control over their learning [70], and they are more satisfied with the teaching [71], even achieving higher levels of self-esteem [72]. Here, as with excessive EA, care must be taken with excessive support for autonomy, especially when it is related to very high emotional support, given that overprotection can also negatively affect self-esteem [73].

## 5. Conclusions

In summary, support for autonomy from the teacher has a direct and positive effect on the dimensions of EA, clarity, and repair, and these in turn affect the students’ positive or negative self-esteem. Specifically, EC and repair have a direct positive effect on positive self-esteem, while EA has a direct positive relationship with negative self-esteem. In this way, one can conclude that, to increase positive self-esteem among students, it is necessary for the teacher to support autonomy in order to increase EC and repair. In contrast, if the focus of support for autonomy is placed on the EA of students, this will increase negative self-esteem. This aspect is relevant for teachers to properly apply AS in PE classes to ultimately improve the positive self-esteem of secondary-school students.

## Figures and Tables

**Figure 1 children-10-01690-f001:**
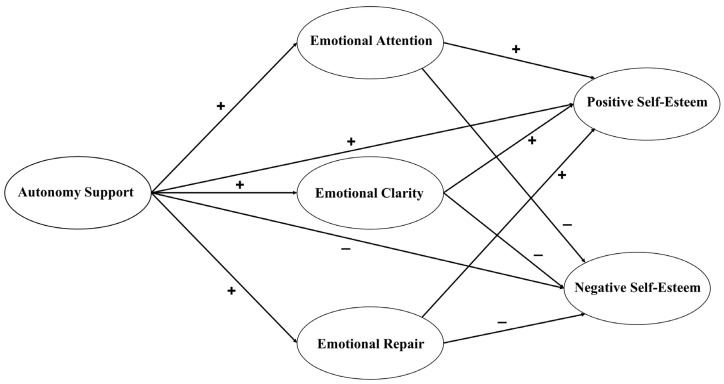
Hypothesized model.

**Figure 2 children-10-01690-f002:**
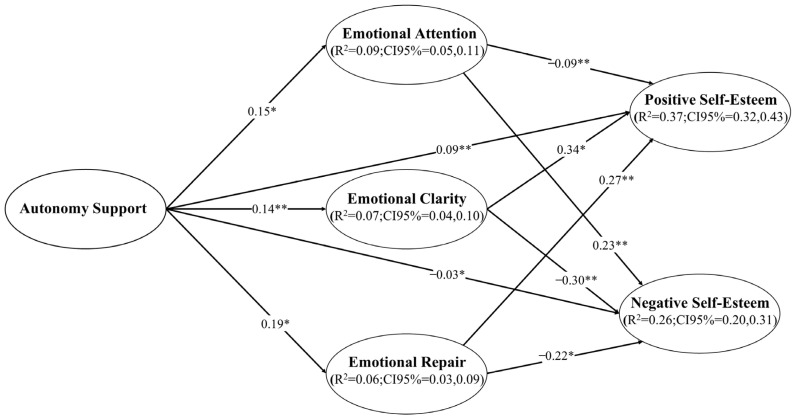
Predictive relationships of autonomy support on positive self-esteem and negative self-esteem through EI mediation. Note: ** *p* < 0.01; * *p* < 0.05. R^2^ = Variance explained; CI = Confidence interval.

**Table 1 children-10-01690-t001:** Descriptive statistics and correlations between variables.

Variable	Range	*M*	*SD*	Q1	Q2	ω	2	3	4	5	6
1. AS	1–7	4.96	1.43	−0.47	−0.56	0.95	0.15 **	0.23 **	0.23 **	0.29 **	−0.13 **
2. EA	1–5	3.26	0.96	−0.19	−0.55	0.88		0.32 **	0.21 **	0.05	0.13 **
3. EC	1–5	3.16	0.87	−0.13	−0.55	0.85			0.53 **	0.41 **	−0.27 **
4. ER	1–5	3.40	0.86	−0.17	−0.58	0.83				0.42 **	−0.31 **
5. +self-esteem	1–4	3.24	0.64	−0.79	0.25	0.80					−0.57 **
6. −self-esteem	1–4	2.16	0.71	0.35	−0.60	0.75					

Note. ** Correlation is significant at the 0.01 level; AS = Autonomy Support; EA = Emotional Attention; ER = Emotional Regulation; EC =Emotional Repair; *M* = Mean; *SD* = Standard Deviation; Q1 = Skewness; Q2 = Kurtosis; ω = McDonald’s Omega; AS = Autonomy Support.

**Table 2 children-10-01690-t002:** Estimation of the significant standardized parameters and statistics of the mediation model.

	Independent Variable	Dependent Variable	Mediator	β	SE	95%CI	
						Inf	Sup
Direct effects							
	AS	EA		15 *	0.04	0.08	0.21
	AS	EC		0.21 **	0.04	0.15	0.28
	AS	ER		0.19 *	0.04	0.11	0.25
	AS	+Self-esteem		0.09 **	0.04	0.06	0.12
	AS	−Self-esteem		−0.03 *	0.04	−0.05	−0.01
	EA	+Self-esteem		−0.09 **	0.04	−0.16	−0.03
	EA	−Self-esteem		0.23 **	0.04	0.16	0.28
	EC	+Self-esteem		0.34 *	0.05	0.25	0.41
	EC	−Self-esteem		−0.30 **	0.05	−0.38	−0.23
	ER	+Self-esteem		0.27 **	0.05	0.20	0.35
	R	−Self-esteem		−0.22 *	0.05	−0.28	−0.13
	Gender	EA		0.45 *	0.07	0.33	0.56
	Gender	EC		−0.26 **	0.06	−0.36	−0.16
	Gender	ER		−0.24 **	0.07	−0.35	−0.14
	Age	EA		0.09 *	0.03	0.05	0.13
	Age	ER		−0.05*	0.02	−0.10	−0.01
	Educational centre	−Self-esteem		0.02 **	0.01	0.01	0.04
Indirect effects							
	AS	+Self-esteem	EA	−0.01 **	0.00	−0.02	0.00
	AS	−Self-esteem	EAA	0.04 **	0.00	0.02	0.06
	AS	+Self-esteem	EC	0.07 **	0.01	0.04	0.09
	AS	−Self-esteem	EC	−0.06 **	0.01	−0.08	−0.03
	AS	+Self-esteem	ER	0.05 **	0.01	0.03	0.07
	AS	−Self-esteem	R	−0.04 **	0.01	−0.06	−0.01
Total indirect effects							
	Gender	+Self-esteem		−0.10 **	0.02	−0.13	−0.06
	Gender	−Self-esteem		0.11 *	0.02	0.08	0.14
	Age	+Self-esteem		0.06 **	0.02	−0.07	−0.01
	Age	−Self-esteem		−0.04 *	0.02	0.03	0.08

Note. β = Estimation of standardized parameters; AS = Autonomy support; EA = Emotional attention; ER = Emotional regulation; EC = Emotional repair; SE = Standard error; 95%CI = 95% confidence interval; Inf = Lower limit of 95%CI; Sup = Upper limit of 95%CI; ** *p* < *0*.01; * *p* < *0*.05.

## Data Availability

The data presented in this study are available on request from the corresponding author. The data are not publicly available due to privacy.

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
