# Peer review of "Effect of a Physical Education Teacher’s Autonomy Support on Self-Esteem in Secondary-School Students: The Mediating Role of Emotional Intelligence"

_children, 2023, doi:10.3390/children10101690_

Round 1
Reviewer 1 Report
Thank you for the opportunity to read this paper.
The paper follows the rules of a good article.
The title is adequate for this research. The aim is clear. The references are representative.
The outline of the research question. The research question makes sense in light of the subject's existing knowledge. The subject selection procedure is transparent. Although the authors said that (R2) is the explained variance in Line 216, which is untrue, the techniques are adequately defined and implemented. is the correlation coefficient, which presumes a connection between the variables. To enable the study to be repeated, the authors provide sufficient detail.
I much like the SEM analysis in the essay since it presents novel and unexpected concepts in the area. The end product avoids repetition and highlights statistically significant information.
I like that the writers included some useful, doable advice.
The findings are covered from a variety of perspectives and put into perspective without being overly interpreted.
The results are discussed from multiple angles and placed into context without being overinterpreted.
The conclusions are based on the aims of the study.
The limitations of the study are important but bring opportunities to inform future research
Overall, the article is consistent within itself.
Author Response
We appreciate this reviewer's positive comments on the complete manuscript and in detail by sections.
Reviewer 2 Report
1)The general objective and specific objectives should appear at the end of the introduction. The objective should be clearly written, referring to the population, the intervention, the comparison and the results (PICO strategy).
2)For each group, the number of participants who received the intended treatment and were analyzed for the primary outcome should be indicated. A table showing the demographic and clinical baseline characteristics of each group should be presented.
3) A more explicit detailing of the inclusion and exclusion criteria for participants would help, as would a flowchart or diagram to better illustrate the experimental procedures and timelines.
4)Discussions/Point out the strengths and limitations of your study.
5)The general objective and specific objectives should appear at the end of the introduction.
6)The sample is very small for this type of study that should pay careful attention to its inference results, and should be limited in the article.
7)it would be helpful to add the validity of the proposed form of the questionnaire.
Author Response
1) The general objective and specific objectives should appear at the end of the introduction. The objective should be clearly written, referring to the population, the intervention, the comparison and the results (PICO strategy).
- Response: The purpose has been modified to added population. But this study is not an intervention and we do not compare groups.
2) For each group, the number of participants who received the intended treatment and were analyzed for the primary outcome should be indicated. A table showing the demographic and clinical baseline characteristics of each group should be presented.
- Response: The authors don’t understand this comment since there are not different groups, it is a cross-sectional study.
3) A more explicit detailing of the inclusion and exclusion criteria for participants would help, as would a flowchart or diagram to better illustrate the experimental procedures and timelines.
- Response: We could include a timeline, but this study is not a pre-post test (or intervention study) and is not a systematic review (in order to include different step with the population selection). Inclusion criteria are included (lines 151-153).
4) Discussions/Point out the strengths and limitations of your study.
- Response: Limitations and future prospects are specified in the Sections 4.1 and 4.2.
5) The general objective and specific objectives should appear at the end of the introduction.
- Response: It has been modified and included.
6) The sample is very small for this type of study that should pay careful attention to its inference results, and should be limited in the article.
- Response: The size sample was calculated, and I think that the sample was enough to do this study, more than 1000 participants (i.e., 1069). You can find information about a priori analysis of the required sample size was carried out in section 2.1 (Design and Participants)
7) it would be helpful to add the validity of the proposed form of the questionnaire.
- Response: It is included in the study with McDonald’s omega.
Reviewer 3 Report
First of all, thank you for the opportunity to review this article!
I suggest a series of improvements that would add value to the article and, consequently, the positive decision to publish the article in the journal:
- I suggest rethinking the title. It is not clear who provides this support for autonomy and PE students are actually students who have included PE classes in the program. If they are students at PE vocational schools, then it should be specified, but this aspect is not clear from the text.
- the introduction of some statistical data in the Abstract
- either in the presentation of the instruments or in the results, it would be useful to introduce the Alpha Cronbach values obtained on each scale and its dimensions to check the degree of validity and fidelity of the results.
- although the latent variables (gender, age, educational center) are announced, no statistical data are included in the Results, nor in the Discussions and Conclusions is there any reference to the statistical differences / influences that these variables could have on emotional intelligence, self-esteem, the support for students' autonomy in PE lessons. The introduction of these results can provide perspectives for comparison with other studies, as well as possible explanations for these findings.
- it would be useful to enter details regarding the content of the PE lessons/PE curriculum, the number of hours, the time during the school year when the testing was carried out, but also clarifications regarding the fact that the subjects do not follow the PE classes with the same professor. Moreover, certain aspects from this perspective can be included in the limits. It is possible that there is a valid general framework regarding the PE curriculum (objectives, content, implementation methods, etc.), just as it is possible that there is a degree of freedom of the teacher/school regarding the PE curriculum with reference to the subject of the article - the development of the skills to work independently and the support from the teacher for this. All these details can influence the current study and should be taken into account in the research process.
-
Author Response
- I suggest rethinking the title. It is not clear who provides this support for autonomy and PE students are actually students who have included PE classes in the program. If they are students at PE vocational schools, then it should be specified, but this aspect is not clear from the text.
- Response: It has been modified according to reviewer suggestion.
- the introduction of some statistical data in the Abstract
- Response: It has been included and modified in the Abstract.
- either in the presentation of the instruments or in the results, it would be useful to introduce the Alpha Cronbach values obtained on each scale and its dimensions to check the degree of validity and fidelity of the results.
- Response: We have included the McDonald’s omega value due to the characteristics of the questionnaire. It has been included in instrument section. If the reviewer considers checking the Alpha’s value we could do it too. Following Ventura-León and Caycho-Rodríguez (2017), alpha's Cronbach has some limitations given that it could be affected by number of the items from the number of response alternatives and the proportion of the test variance (Domínguez-Lara & Merino-Soto, 2015a) and for that reason it is recommended to use of McDonald’s omega.
Ventura-León, J. L., & Caycho-Rodríguez, T. (2017). Cartas al director. El coeficiente Omega: un método alternativo para la estimación de la confiabilidad. Revista Latinoamericana de Ciencias Sociales, Niñez y Juventud, 15(1),625-627. https://www.redalyc.org/pdf/773/77349627039.pdf
Domínguez-Lara, S. A. D. & Merino-Soto, C. M. (2015). ¿Por qué es importante reportar los intervalos de confianza del coeficiente alfa de Cronbach? Revista Latinoamericana de Ciencias Sociales, Niñez y Juventud, 13(2), 1326-1328. https://www.redalyc.org/pdf/773/77340728053.pdf
- although the latent variables (gender, age, educational center) are announced, no statistical data are included in the Results, nor in the Discussions and Conclusions is there any reference to the statistical differences / influences that these variables could have on emotional intelligence, self-esteem, the support for students' autonomy in PE lessons. The introduction of these results can provide perspectives for comparison with other studies, as well as possible explanations for these findings.
- Response: Thank you very much for this comment. The model was controlled by gender, age and educational center to avoid the influence of these variables on the SEM model. Since differences according to these variables were not the objective of this study, they were not included in the results or discussion. However, the significant results have been included in Table 2 and it is included as a future line of research to delve into the differences, especially according to gender and age in emotional intelligence. Now, we think the results are clearer to potential readers.
- it would be useful to enter details regarding the content of the PE lessons/PE curriculum, the number of hours, the time during the school year when the testing was carried out, but also clarifications regarding the fact that the subjects do not follow the PE classes with the same professor. Moreover, certain aspects from this perspective can be included in the limits. It is possible that there is a valid general framework regarding the PE curriculum (objectives, content, implementation methods, etc.), just as it is possible that there is a degree of freedom of the teacher/school regarding the PE curriculum with reference to the subject of the article - the development of the skills to work independently and the support from the teacher for this. All these details can influence the current study and should be taken into account in the research process.
- Response: The information about PE has been added in section 2.1 (Design and Participants). As well, this aspect has been added as a limitation of this study.
Round 2
Reviewer 2 Report
thank you very much for clarifying the doubts after the first reading of the paper. As I said in the first revision, from a reader point of view, I think this study is sound.